# The Identification and Applicability of Regional Brand-Driving Modes for Agricultural Products

**Xiaoping Zheng, Qiuyi Huang and Shuangyu Zheng ***

College of Economics and Management, China Agricultural University, Beijing 100083, China; zhxp@cau.edu.cn (X.Z.); b20193110781@cau.edu.cn (Q.H.)

* Correspondence: s20203111978@cau.edu.cn

**Abstract:** The regional brand-driven construction of agricultural products has taken shape in China. At present, the status quo entails the homogenization of the brand-driven mode of construction, making it a serious phenomenon in China. In addition, the misalignment between the brand-driven mode and resource conditions in some areas not only causes a waste of resources but also leads to a lack of competitiveness and premium capacity for agricultural products within the brand, which cannot increase farmers' income. This article constructs a theoretical model of the brand-driven mode and uses the fuzzy set qualitative comparative analysis method to identify effective brand-driven modes and explore their applicable environmental conditions. This research can provide theoretical guidance for the local development of regional brands of characteristic agricultural products. The results of the driving mode validity analysis show that the four brand-driven modes, resource-dependent, technology-induced, culture-driven, and industry-based, are the main construction paths for regional brands of agricultural products in China. Among them, the effectiveness of the resource-dependent and technology-induced modes is the highest, reaching 0.90 or more. The results of the applicability analysis show that the resource-dependent mode is suitable for farming areas with well-developed supporting policies and infrastructure and good economic development. In addition, the use of the technology-induced mode requires local farmers to have a high level of education and a high-quality base.

**Keywords:** agricultural regional brand; driving mode; applicability; fuzzy set qualitative comparative analysis

## 1. Introduction

As a crucial form of agricultural brand construction, regional brands of agricultural products have received great attention from all walks of life and governments at all levels. In addition, the Central Document No. 1 has emphasized the need to vigorously promote the construction of special agricultural products and their regional brands. Theory and practice have proven that brand building can effectively enhance the international competitiveness of products and form a strong premium effect [1–3]. For example, the Xing'an League in northeastern Inner Mongolia has a long history of growing rice with an annual output of 700,000 tons. Due to the lack of branding, well-known rice producing areas and processing enterprises in the northeast have used 60% of the Xing'an League's stock of low-price high-quality rice. The Xing'an League Executive Office created a regional brand of Xing'an League rice in 2020, which led to an over 30% increase in volume and price for two consecutive years. In recent years, under the active exploration of cultivators such as governments and associations at all levels, the regional brands of agricultural products have developed rapidly in China [4]. This development has cultivated many brand products with a strong influence and high added value, such as Wuchang rice, Yantai apples, and Qingyuan mushrooms. China has gradually standardized the laws on regional

brands of agricultural products, and the number of trademark registrations for brands has increased year after year, from 600,000 in 2008 to 4,812,500 in 2019 [5].

Compared to the economically developed regions in the east such as Zhejiang and Shandong, the brand building in remote mountainous areas in the west with backwards economies and information is not very optimistic. The discovery of a method to build the regional brand for agricultural products in these areas so that farmers can increase their income has become a problem that needs to be studied and solved at this stage. Different industries, species, and regions have different construction paths and methods. In addition, these paths have made useful explorations and attempts in various places, with different construction modes and development paths emerging [6–8]. Due to the short construction time and lack of experience in China, various problems have emerged and have resulted in serious homogenization, backward agricultural production methods, and the insufficient exploitation of cultural resources. This is because of the generalization of the construction mode, which has led to the brand building subjects "learning from each other". They lack the motivation for innovation and ignore the differences in the basic development factors such as natural resources, a humanistic background, and the industrial base of the region where the brand is located. Whether the initial choice of the brand construction mode is reasonable determines, to a certain extent, whether the brand will ultimately be successful. The government or associations and other brand cultivation bodies should concentrate on examining the regional characteristics of crops and cultural and historical situations. In addition, they should cultivate regional brands of agricultural products that are compatible with the characteristics of the regional resources.

Some scholars have briefly classified the driving modes of agricultural regional brands, such as Ma, Q.X. (2010) and Zhang, C.T. (2015), who have proposed various models, including the special-premium natural resource mode, the modern science and technology mode, the local human resource mode, and so on. However, these classifications lack the support of empirical studies and the analysis of the applicability of the driving modes. Therefore, based on the dominant advantageous resources, this paper constructs the theoretical model of the driving mode of agricultural regional brands. In addition, we conduct empirical research on the effectiveness and applicability conditions of the brand-driving mode by a fuzzy set qualitative comparative analysis (fsQCA) method. The purpose of this study is to solve the problems of homogenization and formalization in brand construction, guiding the layout of agricultural regional brands to develop towards specialization, scale, and coordination.

## 2. Literature Review

### 2.1. Regional Brands of Agricultural Products

The term "regional brand of agricultural products" belongs to the characteristic brands proposed by domestic scholars and practitioners based on the experience of foreign agricultural brand building and the actual domestic agricultural development. In addition, it is commonly applied in domestic agricultural branding. Lu, G.Q. (2002) took the lead in applying location branding to the field of agricultural product branding and proposed to develop a regional brand of agricultural products. A regional brand of agricultural products refers to the brand of agricultural products in a region with a unique natural environment, history, and humanity, expressed as "origin name + product (category) name", for example, Wuchang rice and Gannan navel oranges. It is jointly used by the government, associations, and agricultural enterprises [9].

A regional brand of agricultural products is a public good with basic characteristics such as a regional identity and industrial clustering, public (non-competitive and non-exclusive) and external economies, government endorsement, and industry specificity. Its essential characteristics, public and external economy, have an important dual role in facilitating and hindering the management and development of the brand. Once poorly controlled, it is very easy to make a regional brand of agricultural products into the "tragedy

of the commons". Its construction, publicity, or brand maintenance must rely on the participation of the government, industry associations, enterprises, farmers, and other forces [10,11]. Therefore, it is necessary to clarify the different responsibilities and interests of pursuit of each party in the various stages of brand cultivation.

*2.2. The Identification of Brand-Driving Modes*

At present, scholars classify and identify the driving modes of agricultural regional brands through the two perspectives of construction subjects and leading resources, using qualitative methods such as typical case analyses and general descriptions. Under the strong impetus of a particular subject, the regional branding of agricultural products is realized through the coordination and cooperation of multiple parties. There are three main cultivating bodies: local governments, enterprises, and associations [12–14]. Some scholars have proposed two modes for the division of labor with respect to cultivating the regional brands of agricultural products based on the perspective of construction subjects. One of these modes is the "government-led, intermediary role of chambers of commerce and associations, and enterprise participation", and the other is the mode of "government decentralization, chamber of commerce-led, association organization, and enterprise sponsorship" [15]. Both cooperatives and regional brands have attributes and externalities, and both can generate economies of scale; furthermore, combining the two for operation is a feasible mode [16]. With the continuous improvement of agricultural modernization and the industrialization system, the role of leading enterprises in the construction of regional brands of agricultural products has become increasingly prominent. It has led to the formation of a third "leading enterprises-led" mode for the creation of regional brands of agricultural products [17].

In addition, brand cultivation should examine the distribution of regional resources [18] and generally must be built on the basis of unique regional resource industries, which can be formed through the advantages of native resources. Therefore, most scholars have categorized and identified various agricultural regional brand-driving modes in China from dominant resource advantages of specific regions through case observations and multi-case comparative analyses. In addition, there are over ten kinds of various agricultural regional brand-driving modes proposed. These brand-driven modes are similar to the more commonly used driving modes such as the special-premium natural resource mode or the resource endowment-driven mode, the modern science and technology mode or the technology innovation-driven mode, the local human resource mode or the human history inheritance mode, and the industry chain type or industry cluster driven [19–22]. There are only differences in names or callings, all of which focus on the four aspects of natural resources, history and humanity, science and technology, and the market industry for resource mining.

*2.3. The Applicability of Brand-Driven Modes*

The study of the applicability of brand-driven modes is based on sorting out and classifying the existing driving modes of regional brands of agricultural products, exploring the environmental conditions for the applicability of each brand-driven mode. Most of the research literature on the driving modes of agricultural regional brands remains at the first level, only briefly classifying and summarizing the existing driving modes of agricultural regional brands [19–22]. The second level of inquiry into the environmental conditions applicable to each brand-driven model still leaves many gaps. More relevant to the study of the applicability of the brand-driven mode is the research on the meso-environmental elements of the development of agricultural regional brands. The object of the former is agricultural regional brands that have already been categorized, while the object of the latter is agricultural regional brands as a whole. In terms of theoretical logic, the latter study includes the former, while the former extends the latter. In addition, their common content of their research consists of the environmental conditions of brand applicability. Starting from the meso-environmental elements of agricultural regional

brands, we explore the environmental conditions applicable to each brand-driving mode. This exploration is consistent with the research logic of the existing literature and can also effectively fill the research gaps in the driving mode of agricultural regional brands.

## 3. Theoretical Model

At this stage, it is more common to study the brand-driven mode for agricultural products from the perspective of the dominant advantageous resources. This article will also follow this paradigm, and based on a summary of the key elements of brand construction, it will construct a theoretical model of the driving mode of agricultural regional brands from the dominant advantageous resources.

### 3.1. Key Elements of the Brand Construction

Location, industry, and environment are the key elements in the construction of regional brands of agricultural products, among which location is the most important element. It determines the inherent genes for the sustainable development of regional brands of agricultural products. Throughout the theoretical views of scholars on the brand building elements, the locational and industrial elements dominate the creation path and development direction of regional brands of agricultural products. In addition, the environmental elements guarantee the successful implementation of the creation path and development direction.

Locational elements mainly refer to the unique or high-quality resource endowment elements within the geographical area to which a certain brand belongs that differ from those within the geographical area to which other brands belong. In addition, locational elements are reflected in the three aspects of natural resource elements, technical resource elements, and cultural resource elements [23–25]. Industrial elements refer to industries such as product production and processing, brand design, and promotion, which gather in large numbers in a special area, forming industrial scale, industrial modernization, and industrial economic and service integration [21,26]. They are also known as industry clusters, which can produce a radiation-driven effect and are the basis for expanding the regional brand.

Environmental elements refer to the political, economic, legal, and technological macro-environment and the micro-environment composed of competitors in the same industry, the manufacturers of production substitutes, and the suppliers of raw materials [27]. This article divides the macro- and micro-environments into hard and soft environments, as shown in Figure 1.

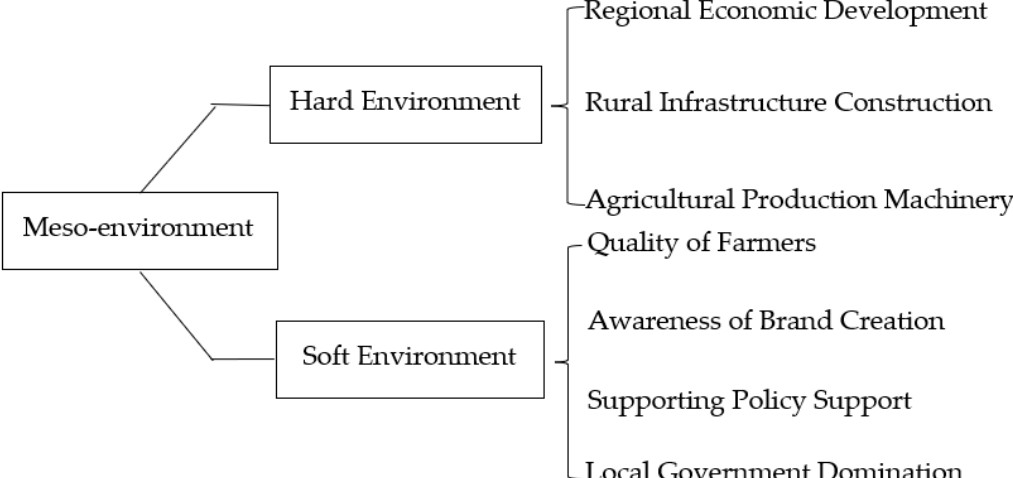

**Figure 1.** Brand-driven meso-environment.

### 3.2. A Theoretical Model of the Brand-Driving Mode

Regional resources, industrial resources, and hard and soft environments affect the construction, management, and development of regional brands of agricultural products at different levels and to different degrees. Regional brands' public goods and other characteristics cause them to face a more complex network of factors than agricultural brands. These factors will not only individually affect the results of the construction of regional brands of agricultural products but will also interact with each other to produce the "substitution effect" or "complementary effect". In addition, they finally work together during the brand's formation, which is the so-called "joint effect", as shown in Figure 2.

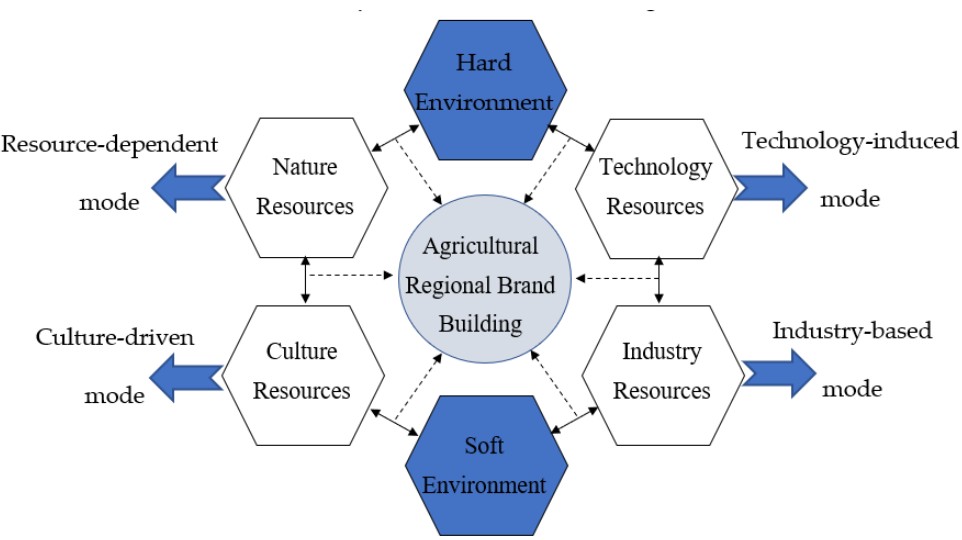

**Figure 2.** Theoretical model of the agricultural product regional brand-driving mode.

On the one hand, this article makes use of the different elements and core advantageous resources in the construction of various agricultural products regional brands to initially identify and classify brand-driving modes, proposing four resource-dependent, technology-induced, culture-driven, and industry-based modes. On the other hand, the effectiveness of a brand-driving mode requires a certain environmental foundation. To better investigate the environmental conditions under which the driving mode produces the best results, this article analyzes the applicability of the driving modes based on the identification of the driving modes. Therefore, the environmental factors (hard and soft environments) in the theoretical model focus on the meso-environment from the selection of the brand-driving mode to the result of the brand construction. In addition, the ultimate success of the regional brand of agricultural products depends on the suitability of the driving mode to the environmental conditions.

### 4. Research Methods

Our research design consisted of the following phases: (1) Study 1, identification and validity analysis of driving modes of regional brands for agricultural products, and (2) Study 2, analysis of the applicability of the brand-driving mode, investigating the environmental conditions under which the driving mode produces the best results.

### 4.1. Fussy-Set QCA

A qualitative comparative analysis, unlike previous statistical analyses or single-case studies, is an intermediate state between the two. In addition, it is more suitable for the analysis of "multiple concurrent causality" at the mid-macro level, i.e., when the sample is limited. After over thirty years of continuous enrichment and development, qualitative

comparative analysis contains three specific methods for different data types, but all are valid, namely QCA, mvQCA, and fsQCA. Among them, the most widely used is QCA, because it is the earliest and simplest method. However, it is too simplistic to only calibrate the data as "1" or "0", which would imply that a variable has only two sides: "yes" or "no". This simple and brutal method of data processing may lead to errors in the results, ignoring the subtle changes in the data. The method of mvQCA is more suitable for discontinuous variable values, such as fixed-order data. The method of fsQCA is a pioneering combination of set theory and qualitative comparative analysis methods, breaking through the traditional division between qualitative and quantitative analysis. The study variables and data in this paper are all dichotomous or continuous variable data, among which continuous variable data are the main data, occupying over 80% of the total data volume. Therefore, this paper focuses on the fsQCA method, which uses the specialized software fsQCA 3.0 to process and run the data. The main steps are as follows:

- Variables—determining the outcome variable and selecting appropriate condition variables. The number of condition variables is controlled in the best range of 6–7.
- Sample—selecting appropriate cases based on the settings of result and condition variables. The number of conditional configurations generated in the data analysis grows exponentially with the number of conditional variables. To avoid the problem of sample finiteness, i.e., too small of a sample resulting in no corresponding cases for the conditional configuration, the number of cases should be no less than the types of conditional configuration.
- Calibration of fuzzy set data—establishing coding criteria to code the outcome and condition variables. The coding criteria entails three anchor points: fully affiliated anchor points, intersection points, and fully unaffiliated anchor points.
- Descriptive statistical analysis and causality test—carrying out a descriptive statistical analysis of the calibration results to show the data distribution of each variable after calibration. In addition, before conducting the necessity and sufficiency analysis, this paper tests the causality of the outcome and condition variables to ensure the validity of the results.
- Analysis of the necessity of individual conditions—testing whether individual conditions (including their non-sets) constitute the necessary conditions for outcome.
- Sufficiency analysis of conditional configuration—running the program. Using Boolean minimization, it extracts the factors and configurations that play a key role in the outcome variable and builds an explanatory model.
- Analysis of the applicability of the driving mode—same as the sufficiency analysis of conditional configuration. However, In the step involving the generation of an intermediate solution, we set the conditions for resource-dependence or technology-induced "presence".

### 4.2. Sample

The sample selection of this paper adheres to the following principles: (1) maximum similarity and maximum heterogeneity. Maximum similarity means that the case situation should include the antecedent variables, i.e., identifying variables and basic variables. Maximum heterogeneity means that the sample selection should include counterfactual cases. (2) The principle of data accessibility. At present, the official detailed data of each agricultural regional brand are limited, complicated, and uneven. We need to fully understand the amount of data available, its availability, and the ease of access of the sample to be selected. Then, we conduct secondary screening of the sample based on meeting the data requirements of the study. (3) The principle of matching the variable settings with the sample size. Due to the multiple concurrencies of events, the occurrence of an outcome may be caused by multiple condition configurations. In addition, the number of condition configurations is exponentially related to the number of antecedent condition variables (number of condition configurations = $2^n$, where n = number of condition variables). The

sample size needs to be larger than the number of condition configurations, otherwise there will be a limited diversity of cases, which will affect the results of the model runs [28].

Based on the above principles, the study selected 40 case samples after preliminary selection and secondary screening after data survey (Table A1). The case samples are from agricultural regional brands in cities or provinces in the eastern, central, and western regions, with economies at developed, medium, and less-developed levels. The samples involve several agricultural product industry categories such as fruit, tea, and grain. The top five provinces with the largest number of brand case distributions are Henan Province, Shandong Province, Zhejiang Province, Shanxi Province, and Sichuan Province. Among them, fruit accounts for 30%, vegetables for 15%, grain for 13%, livestock and poultry for 13%, tea for 10%, Chinese herbs for 8%, edible mushrooms for 8% and aquatic products for 5%. The distribution of provinces and types is basically in line with the status quo of registration of geographical indication certification marks in China and is representative [29].

### 4.3. Variables

One output and four input variables (identification condition variables) were used in Study 1; one output and five input variables (basic condition variables) were used in Study 2. The variables were selected based on the literature review (Figure 1 and Figure 2).

### 4.3.1. Outcome Variable (Brand Effect)

Domestic and international scholars assess the effectiveness of brand building in terms of two aspects: brand value and brand influence [30,31]. Brand effectiveness refers to the ability of a brand to drive consumers' perceptions, evaluations, and behavioral intentions to influence their purchasing decisions, enhancing products' competitiveness and creating a strong price premium. Given that the subject of this study involves several product categories, the brand value of each product category itself varies to a certain extent and cannot be compared uniformly. This study uses the relative brand influence index as a measure of the brand effect of the outcome variable, thus eliminating errors arising from differences in product categories, with data from the "The first batch of brand value assessment and influence index evaluation list of China's agricultural brand directory".

### 4.3.2. Identification Condition Variables

1. Resource-dependent. This paper measures the resource dependence of agricultural regional brands based on the number of years of certification for geographical indication-protected products, geographical indication-certification marks, and geographical indications for agricultural products. The longer the certification time, the greater the resource advantage they have, calculated as RESOURCE = fussyor (GIAP, GICM, and GIPP). Where RESOURCE is resource-dependent, GIAP denotes geographical indication for agricultural products, GICM denotes geographical indication certification mark, and GIPP denotes geographical indication protected product. Fussyor () is the function for establishing macro variables in the QCA method.

2. Technology-induced. This variable combines three micro variables: agricultural technology inputs, agricultural technology promotion, and agricultural technology outputs [32]. We establish the macro variable technology-induced, calculated as TECHNOLOGY = f (ATI, ATP, ATO). Where TECHNOLOGY is technology-induced, ATI is agricultural technology input, ATP is agricultural technology diffusion, and ATO is agricultural technology output. F () is the weighting function for the calibrated fs/QCA data, using an outcome-oriented weighting method of weighting coefficients set to 0.3, 0.3, and 0.4. Agricultural technology input refers to agricultural research expenditure, calculated as average annual research expenditure per hectare of arable land = research and experimental

development expenditure by municipality / actual arable land area at the end of the year, using data published in the Statistical Yearbook. The average annual frequency of agricultural technology training and extension conducted by brand owners was used as a measure of agricultural technology extension. In addition, data were obtained by searching government portals using technical training as a search term. The number of invention patent applications by brand-authorized enterprises is a measure of agricultural technology output, using data from the "2019 China Agricultural Products Regional Brand Catalogue Declaration".

3. Culture-driven. Calculated as CULTURE = f (CR, CP, CI). Where CULTURE is culture-driven, CR is culture resources, CP is culture promotion, and CI is culture innovation. F () is the weighting function for the calibrated fs/QCA data, using an outcome-oriented weighting method of weighting coefficients set to 0.3, 0.3, and 0.4. Culture resources are measured by the possession of local intangible cultural heritage, agricultural cultural heritage, and historical and cultural inheritance. In addition, three types of cultural resources directly related to or available for the regional brand of agricultural products were assigned a value of 3, in decreasing order, and none of them were assigned a value of 0. Culture promotion is measured by the type of cultural communication channels the brand has. In addition, four cultural communication channels—museums, exhibition halls, experience halls, and industry-related large-scale festivals—were assigned a value of 4, in decreasing order, and none of them were assigned a value of 0. Culture innovation involves packaging, advertising, promotion, or festivals, and was assigned a value of 4 if the four types of cultural innovation are present in decreasing order, or 0 if there were none.

4. Industry-based. In this paper, the three micro variables, namely, production scale, leading enterprises, and market share [33,34], are used to establish the macro variable "industry-dependent", which is calculated as INDUSTRY = f (PS, LE, MS). When studying the scale of agricultural operations in China, the actual arable land area is used to show the scale of production. China divides leading enterprises into national, provincial, and municipal levels, assigning them values of 3, 2, and 1, and calculates the total value of authorized enterprises for the regional brand as a measurement indicator [35,36]. In this paper, we calculated market share by applying the basic formula of sales volume of agricultural products of the regional brand / total sales volume of agricultural products of the category to which it belongs.

### 4.3.3. Basic Condition Variables

We have selected the five most important environmental conditions in the process of regional brand construction for agricultural products:

## 5. Results and Discussion

### 5.1. Calibration of Fuzzy Set Data

Through existing theoretical and empirical knowledge, this paper converted the data to fuzzy set affiliation scores using the direct calibration method [42,43] based on the type of data for each condition and outcome. Specifically, there are 680 data points for 17 variables (1 outcome variable and 16 condition variables) and Table 1 summarizes the calibration information for these variables. The fuzzy set data were calibrated primarily based on three qualitative anchors, either manually or using the fs/QCA program. The main methods for setting the three anchor points were as follows: using 95%, 50%, and 5% quantile values as thresholds for full affiliation, the crossover point, and full disaffiliation, respectively [44]; setting the three anchor points based on case-related information, sample data distribution, and empirical knowledge [45]; and the data type was similar to the 5-point ordinal scale, with a score of 4 being fully affiliated, a score of 0 being fully unaffiliated, and 2 being the crossover point [46].

**Table 1.** Calibration of result to conditions.

| Result and Conditions | Calibration | | |
| :---: | :---: | :---: | :---: |
| | **Full Affiliation** | **Crossover Point** | **Full Disaffiliation** |
| EFECT | 87.75 | 74.87 | 61.36 |
| GIAP | 10 | 3 | 0 |
| GICM | 19 | 7 | 0 |
| GIPP | 17 | 7 | 0 |
| ATI | 15,000 | 6000 | 500 |
| ATP | 8 | 4 | 0 |
| ATO | 10 | 4 | 0 |
| CR | 3 | 2 | 0 |
| CP | 4 | 2 | 0 |
| CI | 4 | 2 | 0 |
| PS | 300 | 36.07 | 2 |
| LE | 45 | 12 | 0 |
| MS | 15 | 4.60 | 0.20 |
| GL | 1 | / | 0 |
| PS | 1 | / | 0 |
| RE | 79,605.19 | 25,720.63 | 6570.48 |
| IC | 200 | 155.87 | 50 |
| FQ | 286,743 | 170,677 | 100,000 |

*5.2. Descriptive Statistical Analysis and Causality Test*

As the calibration results of the data in this study are relatively large and cannot be directly displayed in this article, this section will carry out a descriptive statistical analysis of the calibration results (Table A2). The maximum values of both the results and the condition variables after the calibration are close to 1, the minimum values are close to 0. In addition, the mean values are close to 0.5, and the standard vertebral error values are relatively small. This shows that the data distribution of each variable after calibration is relatively even, which can minimize the errors that may arise during the following necessity and adequacy analyses.

Before conducting the necessity and sufficiency analysis, this paper tested the causality of the outcome and condition variables to ensure the validity of the results. The results of the specific analysis are shown in Table 2, whereas the causality plot between the condition variable fRESOURCE and the outcome variable fEFECT is shown in Figure 3 (the causality plots between the other variables are omitted). The causality of the variables in fsQCA was tested by consistency criteria, and the results showed that there is a causal relationship between the condition and outcome variables.

**Table 2.** Results of the causality test.

| X Collection | Y Collection | Consistency $(X \leq Y)$ | Consistency $(X \geq Y)$ |
| :---: | :---: | :---: | :---: |
| fRESOURCE | fEFECT | 0.8040 | 0.7540 |
| fTECHNOLOGY | fEFECT | 0.9018 | 0.6553 |
| fCULTURE | fEFECT | 0.8961 | 0.7464 |
| fINDUSTRY | fEFECT | 0.9570 | 0.6257 |
| fGL | fEFECT | 0.6417 | 0.5604 |
| fPS | fEFECT | 0.6761 | 0.7187 |
| fRE | fEFECT | 0.7754 | 0.7954 |
| fIC | fEFECT | 0.7734 | 0.4822 |
| fFQ | fEFECT | 0.7596 | 0.5027 |

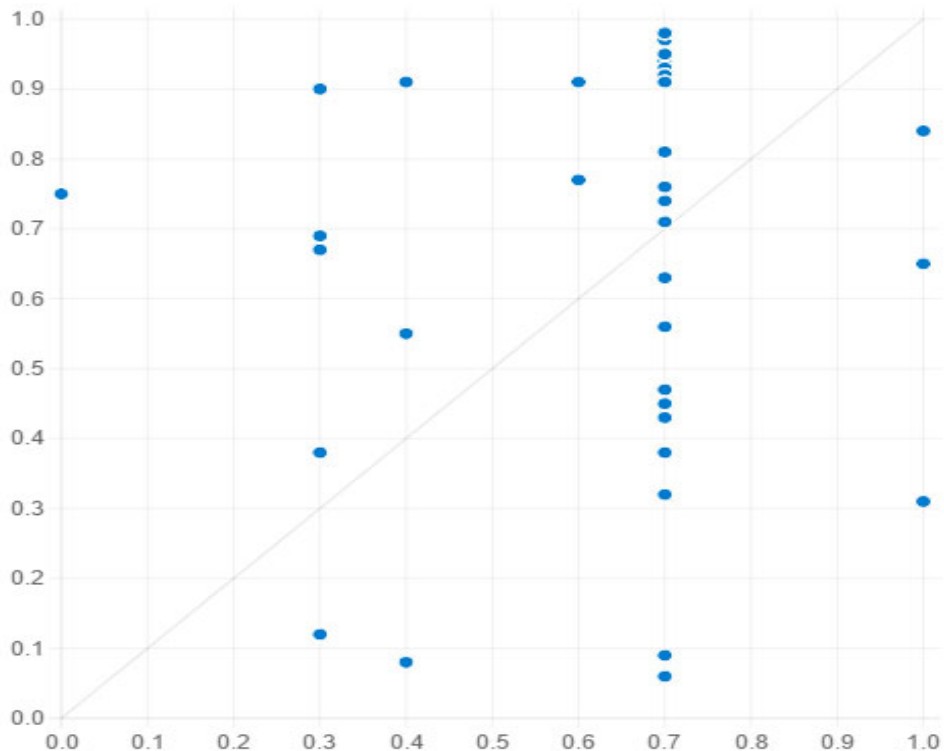

**Figure 3.** Causality diagram for condition variable fRESOURCE and outcome variable fEFFCT.

*5.3. Study 1: Identification and Validity Analysis of Driving Modes*

5.3.1. Analysis of the Necessity of Individual Conditions

In line with mainstream QCA research, this paper first tested whether individual conditions (including their non-sets) constitute the necessary conditions for a strong brand effect. Consistency is an important measure of a necessary condition, and a condition is necessary for the outcome when the level of consistency is greater than 0.9 [47]. The results of the test are shown in Table 3, where the consistency level for all conditions was not higher than 0.9 and was non-essential.

**Table 3.** Analysis of the necessary conditions for a strong brand effect.

| Antecedent Conditions | Strong Brand Effect | | Antecedent Conditions | Strong Brand Effect | |
|---|---|---|---|---|---|
| | Consistency | Coverage | | Consistency | Coverage |
| RESOURCE | 0.8869 | 0.7456 | GL | 0.5604 | 0.6417 |
| resource | 0.2847 | 0.8651 | gl | 0.4396 | 0.6812 |
| TECHNOLOGY | 0.6553 | 0.9018 | PS | 0.7187 | 0.6761 |
| technology | 0.6029 | 0.7613 | ps | 0.2813 | 0.6175 |
| CULTURE | 0.7464 | 0.8961 | RE | 0.7954 | 0.7754 |
| culture | 0.5414 | 0.7896 | re | 0.4009 | 0.8136 |
| INDUSTRY | 0.6257 | 0.9570 | IC | 0.4822 | 0.7734 |
| industry | 0.6591 | 0.7621 | ic | 0.7399 | 0.8265 |
| FQ | 0.5027 | 0.7596 | fq | 0.6902 | 0.8055 |

Note: Uppercase letters indicate the presence of a condition, lowercase letters indicate its absence.

5.3.2. Sufficiency Analysis of Conditional Configuration

This paper finalizes a consistency threshold of 0.86 and a frequency threshold of 1 after considering the following three best practices for the application of the QCA method: first, the truth table rows (configurations) that result in both 0 and 1 should be covered and roughly balanced [48]; second, the frequency threshold should be set to include at least 75% of the observed cases [49]; and third, possible simultaneous subset relationships should be avoided [47]. The fsQCA outputs three types of solutions of varying complexity: complex, parsimonious, and intermediate solutions. In line with the existing research, this paper reports the intermediate solution [44], supplemented by the parsimonious solution. Table 4 shows the results of the configuration analysis of the four conditions forming the configurations (paths) for establishing the effect of regional branding on agricultural products.

**Table 4.** Configurations of conditions for establishing a strong brand effect.

| Antecedent Conditions | Solution | | | |
|---|---|---|---|---|
| | 1 | 2 | 3 | 4 |
| Resource-dependent | ● | | ● | ● |
| Technology-induced | | ⊗ | ● | |
| Culture-driven | | ⊗ | | ● |
| Industry-based | ● | ● | | |
| Consistency | 0.9578 | 0.8946 | 0.9099 | 0.8963 |
| Original Coverage | 0.5858 | 0.2255 | 0.6059 | 0.6921 |
| Unique Coverage | 0.0326 | 0.0171 | 0.0395 | 0.0862 |
| Consistency of Overall Solution | 0.8738 | | | |
| Coverage of Overall Solution | 0.7790 | | | |

Note: ● = core condition present, ⊗ = core condition absent, ● = auxiliary condition present, ⊛ = auxiliary condition absent, and "space" indicates that the condition may be present or absent. Robustness test: adjust the consistency level threshold to judge the difference in the state of the set relationship and the fitted parameters for different configurations.

As evident from the table above, the core condition for Configuration 1 is resource-dependent, for Configuration 2 it is industry-based, for Configuration 3 it is technology-induced, and for Configuration 4 it is culture-driven. According to the principle that the core condition of the conditional configuration is the driving factor of agricultural regional brands, this paper regards configuration 1: fRESOURCE * fINDUSTRY as the resource-dependent agricultural regional brand-building path, configuration 2: fINDUSTRY * ~fTECHNOLOGY * ~fCULTURE as the industry-based agricultural regional brand-building path, configuration 3: fTECHNOLOGY * fRESOURCE as the technology-induced agricultural regional brand-building path, and configuration 4: fCULTURE * fRESOURCE as the culture-driven agricultural regional brand-building path.

From a cross-sectional perspective between the configurations, the consistency levels of condition configurations 1, 2, 3, and 4 were 0.9578, 0.8946, 0.9099, and 0.8963, respectively. Condition configuration 1 has the highest consistency level, condition group 3 the second highest, and condition configuration 2 and condition configuration 4 have the lowest consistency levels. The original coverage of condition configurations 1, 2, 3, and 4 were 0.5858, 0.2255, 0.6059, and 0.6921, respectively. Condition configurations 3 and 4 have the highest original coverage, condition configuration 1 the second highest, and condition configuration 2 the lowest. The results show that the resource-dependent, technology-induced, and culture-driven modes are mainstream in brand building and cultivation. When planning to create regional brands of agricultural products with local characteristics, regional governments have fully recognized the long-term effectiveness of the culture-driven mode in establishing strong brand effects. In addition, they have planned a layout that combines the technology-induced and culture-driven modes for regional

brands of agricultural products. However, because of a relatively short period and a lack of experience, they have not yet been fully effective.

Longitudinally, configuration 1: fRESOURCE * fINDUSTRY shows that the growing scale of industries in the region and the concentration of related industries are secondary conditions for the development of resource-dependent agricultural regional brands. Configuration 2: fINDUSTRY *~fTECHNOLOGY *~fCULTURE indicates that in the absence of the technological and cultural advantages of regional agricultural products, one can choose to establish industry-based regional brands of agricultural products, where regional natural resources and geographical environmental conditions have less of an influence on them. Configuration 3: fTECHNOLOGY * fRESOURCE and configuration 4: fCULTURE * fRESOURCE show that even technology-induced and culture-driven regional brands of agricultural products cannot ignore the importance of regional natural resources and geographical conditions. When building local brands, local governments or industry associations must not detach themselves from reality and solely emphasize the intangible cultural effects of the brand but must base themselves on product quality control and put brand building into practice.

In addition, the overall results of the adequacy analysis of the conditional configuration of Establishing a Strong Brand Effect showed that the consistency of the overall solution was 0.8738 and the coverage of the overall solution was 0.7790, which met the consistency and coverage tests. The above research results show that the theoretical model of the driving mode of regional brands of agricultural products constructed in this paper has passed the empirical test, and the model actually contains 31 brand cases (belonging to the set of four conditional configurations). The specific distribution of the four configurations (driving modes) is shown in Figure 4, which is basically in line with the theoretical results and the real situation.

| Configuration 2: Industry-based | | Configuration 1: Resource-dependent | |
|---|---|---|---|
| Jinxiang Garlic | (0.79,0.84) | Jiu San Soybeans | (0.70,0.77) |
| Anyue Lemon | (0.70,0.94) | Luochuan Apples | (0.58,0.91) |
| Gannan Navel Orange | (0.70,0.95) | Wuchang Rice | (0.58,0.98) |
| Pinggu Peach | (0.67,0.92) | Dali Winter Jujube | (0.57,0.93) |
| Qianjiang Lobster | (0.62,0.95) | Lingbao Apples | (0.56,0.74) |
| Tengzhou Potato | (0.60,0.91) | Cangxi Red Heart Kiwi | (0.53,0.38) |
| Meixian Kiwi | (0.58,0.81) | Wenshan Panax Ginseng | (0.51,0.98) |
| Nanjiang Yellow Sheep | (0.56,0.45) | | |
| Zigui Navel Orange | (0.74,0.65) | Anji White Tea | (0.83,0.97) |
| Tongjiang Silver Fungus | (0.72,0.63) | Luan Melon Piece | (0.75,0.76) |
| Haimen Goat | (0.70,0.31) | Xinyang Mao Jian | (0.70,0.92) |
| Pingquan Shiitake Mushroom | (0.66,0.81) | Cixi Yangmei | (0.67,0.56) |
| Yuyao Squash | (0.65,0.71) | Dongting Mountain Biluochun | (0.65,0.32) |
| Suizhou Shiitake Mushroom | (0.59,0.91) | Hongze Lake Hairy Crab | (0.60,0.69) |
| Yandang Mountain Dendrobium | (0.56,0.90) | Wenxian Iron Stick Yam | (0.55,0.47) |
| Changfeng Strawberry | (0.54,0.55) | Chang Yi Ginger | (0.52,0.38) |
| Configuration 3: Technology-induced | | Configuration 4: Culture-driven | |

**Figure 4.** Distribution of brand samples on the four configurations (driving modes). Note: cases include (affiliation in the set of conditions; affiliation in the set of results).

*5.4. Study 2: Analysis of the Applicability of the Driving Mode*

5.4.1. Conditions of Application of Resource-Dependent Driving Mode

This study establishes the model fEFECT = f(fRESOURCE, fGL, fPS, fRE, fIC, fFQ), taking resource-dependence, local government dominance, supporting policy support, regional economic development, infrastructure construction, and farmer quality as explanatory variables, and the brand effect as the dependent variable. The data of the relevant variables were imported into fs/QCA for an adequacy analysis of the conditional configurations, and the results are shown in Table 5. Based on the above three best practice criteria, this paper determines a consistency threshold of 0.85 and a frequency threshold of 1. In the step involving the generation of the intermediate solution, we set the condition of resource-dependence to "presence" (Assumptions: fRESOURCE (present)), where the simple solution shows that fPS, fRE, and fIC are the core conditions.

**Table 5.** Conditions of application of resource-dependent mode.

| Resource-Dependent | Conditional Configuration | Consistency | Raw Coverage | Unique Coverage |
|---|---|---|---|---|
| Configuration 1 | fRESOURCE * fPS *~fIC *~fFQ | 0.9155 | 0.4199 | 0.1044 |
| Configuration 2 | fRESOURCE * fGL * fPS * fRE * fIC | 0.8788 | 0.2736 | 0.0324 |
| Configuration 3 | fRESOURCE *~fFQ * fRE *~fIC | 0.8313 | 0.1272 | 0.0238 |
| Configuration 4 | fRESOURCE *~fGL * fPS * fIC | 0.8889 | 0.2894 | 0.0672 |

From the above table, we can see that the consistency and coverage of configuration one are higher than other configurations in the resource-dependent driving mode construction path, especially since the coverage reaches 0.42, which is much higher than the other configurations. This indicates that Configuration 1 can maximize the utility of the resource-dependent mode, and that it is possible to build agricultural regional public brands and establish strong brand effects by using the resource-dependent mode under such environmental conditions. In addition, the construction path meets the environmental conditions of some regions and has been popularized and used on a larger scale. Configuration 1: fRESOURCE * fPS *~fIC *~fFQ shows that the resource-dependent brand-driven mode applies to areas where the level of urbanization is not high, but the agricultural preservation is relatively perfect; where the local government is very supportive of the construction of resource-dependent agricultural regional brands; and where the relevant supporting measures are more perfect. The simple solution illustrates that regional economic development (fRE) and infrastructure construction (fIC) also play an important role in resource-dependent branding, while local government-led development (fGL) plays a minor role and may even play a hindering role.

5.4.2. Conditions of Application of Technology-Induced Driving Mode

This study establishes the model fEFECT = f(fTECHNOLOGY, fGL, fPS, fRE, fIC, fFQ), with technology-induced, local government-led, supporting policy support, regional economic development, infrastructure construction, and farmer quality as the explanatory variables, and the brand effect as the dependent variable. We repeat the above steps (Table 6). Based on the above three best practice criteria, this paper determines a consistency threshold of 0.88 and a frequency threshold of 1. In the step involving the generation of an intermediate solution, we set the condition of technology-induced to "presence" (Assumptions: fTECHNOLOGY (present)), where the simple solution shows that fPS and fFQ are the core conditions.

**Table 6.** Conditions of application of technology-induced mode.

| Technology-Induced | Conditional Configuration | Consistency | Raw Coverage | Unique Coverage |
|---|---|---|---|---|
| Configuration 1 | fTECHNOLOGY * fPS *~fIC * fFQ | 0.9915 | 0.3094 | 0.0828 |
| Configuration 2 | fTECHNOLOGY * fGL * fRE *~fIC | 0.9671 | 0.2456 | 0.0672 |
| Configuration 3 | fTECHNOLOGY * fPS * fRE * fRI * fFQ | 0.9605 | 0.2380 | 0.0501 |

From the above table, we can see that there are three main paths for the construction of a technology-induced driving mode. Among them, the consistency and coverage of Configuration 1 are higher than those of Configuration 2 and Configuration 3. Combining the intermediate and simple solutions, it was found that supporting policy support (fPS) and farmer quality (fFQ) have the greatest influence on the construction of technology-induced agricultural regional brands. In addition, their importance is the highest. To successfully establish a technology-induced brand and establish a strong brand effect, the

local government must provide strong support in terms of policies and increase production and processing technology investment in the agricultural sector. Furthermore, they should encourage brand-related subjects including industry associations and leading enterprises to continuously improve the production and processing technology of agricultural products. These steps could provide a good policy environment for technological research, development, and innovation. In addition, local government-led (fGL) and regional economic development (fRE) also have an important impact on technology-induced brand building but are slightly less important than supporting policy support (fPS). Rural infrastructure construction (fIC) has little impact on technology-induced brand building and development and is the least important.

## 6. Conclusions

### 6.1. Findings from Study 1

Resource-dependent, technology-induced, culture-driven, and industry-based brand-driven modes are the four most effective driving modes used for regional agricultural brand building in China. The construction of regional brands of agricultural products in China has the path features of multiple concurrency and causal asymmetry. The four brand-driven modes proposed in the theoretical model can cover about 78% of the brand cases, which are the main construction paths of agricultural regional brands in China and can effectively establish strong brand effects. The effectiveness of the resource-dependent and technology-induced brand-driven modes is the highest among these four modes, which indicates that the resource-dependent and technology-induced brand-driven modes are more effective than the other two modes in terms of maintaining vitality in the long run, establishing and continuously enhancing the brand effect of agricultural regional brands, and ensuring the successful construction and cultivation of agricultural regional brands.

### 6.2. Findings from Study 2

The brand-driven modes differ in their applicable environmental conditions. The core environmental conditions for the resource-dependent regional brand of agricultural products are supporting policy support, regional economic development, and infrastructure construction. The core environmental conditions for the technology-induced regional brand of agricultural products are supporting policy support and the quality of the farmers (human capital exploitation). The two most effective brand-driven modes, resource-dependent and technology-induced, require very different environmental conditions. The resource-dependent mode emphasizes the improvement and modernization of infrastructure. However, the technology-induced mode emphasizes the improvement of the farmers' quality or human capital development to promote the diffusion of production technologies and the improvement of learning capabilities.

### 6.3. Limitations and Future Research

Similar to all empirical studies, this research has certain limitations. First, the inability of conducting field research and the small amount of officially published information related to brand cases created difficulties in obtaining the sample data. In this paper, we select only representative brand cases from some regions. In future research, we can expand the sample size and conduct field research. By further analyzing the differences of the brand-driven modes in each region and industry, the empirical findings of this paper will be more general and rigorous. Second, in Study 2, we only discussed the applicability of four different brand-driven modes in five environmental conditions, such as local government domination and supporting policy support, without considering other conditions in the environment. In future research, we can further add other environmental conditions and behavior conditions of brand-related stakeholders such as authorized companies and research institutions to the applicability analysis of brand-driven modes to build

a more complete research model. Third, the theoretical model of brand-driven modes established in this article is mainly for the special situation of China. We have not studied its applicability to other countries or regions yet. In the future, we can explore the applicability of the model and improve it by further considering the construction of regional brands of agricultural products in other countries or regions to improve the universality of the theoretical model.

**Author Contributions:** Conceptualization, X.Z., Q.H. and S.Z.; methodology, X.Z., Q.H. and S.Z.; software, S.Z.; validation, X.Z. and S.Z.; formal analysis, X.Z.; investigation, X.Z., Q.H. and S.Z.; resources, X.Z., Q.H. and S.Z.; data curation, X.Z., Q.H. and S.Z.; writing—original draft preparation, X.Z., Q.H. and S.Z.; writing—review and editing, X.Z., Q.H. and S.Z.; visualization, S.Z.; supervision, X.Z.; project administration, X.Z.; funding acquisition, X.Z. All authors have read and agreed to the published version of the manuscript.

**Funding:** The APC was funded by the Youth Fund for Humanities and Social Sciences Research of the Ministry of Education of China, grant number No. 19YJC630231; the Beijing Social Science Foundation of China, grant number No. 21JCC099; China Agriculture Research System of MOF and MARA, grant number CARS38.

**Institutional Review Board Statement:** Not applicable.

**Informed Consent Statement:** Not applicable.

**Data Availability Statement:** Not applicable.

**Acknowledgments:** We thank the anonymous reviewers who provided valuable comments on the manuscript.

**Conflicts of Interest:** The authors declare no conflict of interest.

## Appendix A

**Table A1.** Information of brand sample.

| Brand Name | Province/City | Industry Category | Influence Index |
|---|---|---|---|
| Anyue Lemon | | Fruits | 86.36 |
| Nanjiang Yellow Sheep | | Livestock | 73.98 |
| Cangxi red heart kiwifruit | Sichuan | Fruits | 72.65 |
| Tongjiang silver fungus | | Edible Mushroom | 77.16 |
| Fuping Milk Goat | | Livestock and Poultry | 77.82 |
| Meixian kiwifruit | Shaanxi | Fruits | 81.17 |
| Luochuan Apple | | Fruit | 84.90 |
| Dali Winter Jujube | | Fruits | 85.81 |
| Xinyang Mao Jian | | Tea | 85.36 |
| Wenxian Iron Stick Yam | | Vegetables | 74.38 |
| Lingbao Apple | Henan | Fruits | 79.35 |
| Xinxiang Wheat | | Grain | 77.94 |
| Jiaxian Red Bull | | Livestock | 63.91 |
| Suizhou Shiitake Mushroom | | Edible mushroom | 84.82 |
| Qianjiang lobster | Hubei | Aquaculture | 87.75 |
| Zigui navel orange | | Fruits | 77.48 |
| Anji white tea | | Tea | 90.12 |
| Yuyao squash | Zhejiang | Vegetables | 78.62 |
| Cixi plum | | Fruits | 75.93 |

| | | | |
|---|---|---|---|
| Yandang Mountain Dendrobium | | Chinese herbs | 84.24 |
| Wenshan Panax noto-ginseng | Yunan | Chinese herbs | 91.17 |
| Zhaotong Apple | | Fruits | 64.25 |
| Jinxiang Garlic | | Vegetables | 81.94 |
| Tengzhou potato | Shandong | Grain | 84.79 |
| Zhangqiu Onion | | Vegetables | 85.65 |
| Changyi Ginger | | Vegetables | 72.75 |
| Wuchang Rice | Heilongjiang | Grain | 91.01 |
| Jiusan Soybean | | Grain | 80.05 |
| Pingquan Shiitake Mushroom | Hebei | Edible Mushroom | 81.00 |
| Yutian Baotian Cabbage | | Vegetables | 62.83 |
| Haimen goat | | Livestock | 71.30 |
| Hongze Lake hairy crab | Jiangsu | Aquaculture | 78.33 |
| Dongting Mountain Biluochun | | Tea | 84.78 |
| Liuan Gua Pieces | | Tea | 79.71 |
| Changfeng Strawberry | Anhui | Fruits | 75.81 |
| Huoshan Dendrobium | | Chinese herbs | 73.59 |
| Gannan Navel Orange | | Fruits | 87.39 |
| Ningdu Yellow Chicken | Jiangxi | Livestock | 74.30 |
| Wannian Tribute Rice | | Grain | 65.81 |
| Pinggu Peach | Beijing | Fruits | 79.50 |

**Table A2.** Results of descriptive statistical analysis.

| Variables | Mean Values | Standard Error | Minimum Values | Maximum Values |
|---|---|---|---|---|
| fEFECT | 0.6585 | 0.2747 | 0.0600 | 0.9800 |
| fRESOURCE | 0.6175 | 0.2084 | 0.0000 | 1.0000 |
| fTECHNOLOGY | 0.4785 | 0.2409 | 0.0500 | 0.9100 |
| fCULTURE | 0.5485 | 0.2102 | 0.0700 | 0.9700 |
| fINDUSTRY | 0.4305 | 0.2102 | 0.0700 | 0.9400 |
| fGL | 0.5750 | 0.4943 | 0.0000 | 1.0000 |
| fPS | 0.7000 | 0.4583 | 0.0000 | 1.0000 |
| fRE | 0.6755 | 0.2516 | 0.0800 | 1.0000 |
| fIC | 0.4105 | 0.3191 | 0.0300 | 1.0000 |
| fFQ | 0.4358 | 0.3435 | 0.0000 | 0.9500 |

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
