# Peer review of "The Identification and Applicability of Regional Brand-Driving Modes for Agricultural Products"

_agriculture, doi:10.3390/agriculture12081127_

Round 1

Reviewer 1 Report

Necessary corrections:

1.     There is an urgent need to broke too long sentences into smaller ones, preferably with a single verb. For example in lines: 28-30, 31-34, 41-44, 133-138, etc.

2.     There is an urgent need to give vivid examples of the analyzed theory and practice (lines: 31-34).

3.     I please authors to pay enough attention to the citations. Look in the in lines: 34, 37, 40, 49, 53, 133, 140, 260. The citations are not done properly. What does a quotation like “13” in a line 34 mean?

4.     The careful English proofreading and editing must be done! I do not understand some of the sentences. The are not written in the correct, scientific English language. Look at the lines: 34-40.

5.     (…) “Although some scholars” (lines: 63) –  which scholars? It needs to be explained, enrichened and give the authors’ own opinion on it!

      (…) most of the researchers” (lines: 109) – which researchers?

6.     Many sentences should be re-thought. Some of the words like: mainly, partially, however, can be erased. Look at the lines: 74, 109, etc.

7.     (…) Location elements ..... (line: 138) should start the next paragraph.

8.     What do the authors understand under the phenomenon “effectiveness” – it needs an explanation (line: 276).

9.     Please, explain why the regions in the east (lines: 41-44) have achieved fruitful results in the construction of the regional brands for agricultural products?

10.  The Abstract should convince a reader that he/she made a good choice reading the text. An urgent need to build up the body of the Abstract, which is: create sentences regarding the Introduction, Materials and Methods, Results, etc.

11.  Why this model has been chosen? 

- what is the basic foundation for the research: methodological, organizational, etc.

- what are the restrictions of these methods,

- what are the criteria for selecting these methods have been used,

- what are the criteria for selecting the research population have been used,

- and the cited literature as a background for the research.

12. "The relative indicator brand influence index” needs to be explained (lines: 279-280). In my opinion it should be named as: “the relative brand influence index”. I think that "index" and "indicator" are unnecessarily used twice.

12. In order to follow the train of thought, the detailed research methodology should be presented in a table.

13. In order to follow the train of thought, the explanation of the results should be presented.

14. The names of the regional brands from the Table 4 (lines 527-528) should appear earlier in the text.

15. Please transform conclusions directly resulting from the analyses of statistical methods into conclusions of an applied nature.

16. What are the first and second order conclusions of the analyses carried out.

17. Where, in the conclusion, is it proven that hypotheses have been confirmed or rejected.

18. What are the authors’ suggestions to reduce the limitations of the study in order to continue it.

19.  The literature needs to be enrichened, expanded, and updated.

I advise the authors to place the article on:

ZieliÅ„ska-Chmielewska, A.; Mruk-Tomczak, D.; Wielicka-Regulska, A. Qualitative Research on Solving Difficulties in Maintaining Continuity of Food Supply Chain on the Meat Market during the COVID-19 Pandemic. Energies 202114, 5634. https://doi.org/10.3390/en14185634

20.  I advise the authors to use Appendix to order the more important text in the main course of the manuscript (lines: 397-398, Table 2).  

Reviewer 2 Report

(1)  Some of the sentences (upto 70 words in a single sentence) are very long. So, a long sentence may be split into two or three short sentences wherever possible without losing the appropriate meanings. Some sentences are given below. Please check other sentences also.

(a) Therefore, based on the dominant advantageous resources, this paper constructs the theoretical model of the driving mode of agricultural regional brands, and conducts empirical research on the effectiveness and applicability conditions of the brand driving mode by the fuzzy set qualitative comparative analysis (fsQCA) method to solve the problems of homogenization and formalization in brand construction, guiding the layout of agricultural regional brands to develop towards specialization, scale and coordination.

(b) Some scholars have proposed two division of labor modes for cultivating regional brands of agricultural products based on the perspective of construction subjects, one is the mode of “government-led, intermediary role of chambers of commerce and associations, and enterprise participation”, and the other is the mode of “government decentralization, chamber of commerce-led, association organization, and enterprise sponsorship”12.

(c) The brand-driven modes differ in their applicable environment conditions. The core environment conditions for resource-dependent regional brand of agricultural products are supporting policy support, regional economic development and infrastructure construction; the core environment conditions for technology-induced regional brand of agricultural products are supporting policy support and quality of farmers (human capital exploitation).

(d) The two most effective brand-driven modes, resource-dependent and technology-induced, require very different environment conditions, with the resource-dependent mode emphasizing the improvement and modernization of infrastructure, while the technology-induced brand-driven mode emphasizes the improvement of farmer quality or human capital development to promote the diffusion of production technologies and the improvement of learning capabilities.

(2) When we construct a theoretical model, it should be in general nature, applicable to all countries or regions, not only to China. The article constructs a theoretical model of the brand driving mode with respect to China. How will your theoretical model be applicable to any country? Some writings should be given.

(3) Proper English language corrections need to be done. For example, in section 4.3.3, the first sentence “We mainly selects the five more important environment conditions …….. for agricultural products” may be written as “We mainly select or we mainly have selected the five more important environment conditions …….. for agricultural products”

(4) The full article requires proper editing and proper arrangements of items. For example, in Table 5, why black dots and cross dots are coming outside Table?

(5) Analysis and discussion in section 5 should be improved. Analysis is mainly concentrating on own results. Authors may cite references of other studies.  

(6) Every research study suffers from some limitations. What are the limitations of this present study? Authors should mention some of them.

……….x…………

Reviewer 3 Report

Thank you for giving me this opportunity to review this manuscript entitled ‘The identification and applicability of regional brand driving modes for agricultural products.”

Introduction

The contents are not cited at all or the brackets mark is not shown in the manuscript.

The explanation of ‘brand’ and ‘brand driving mode’ should be briefly explained and be supported by previous research in introduction.

Literature review

3.1 Key elements of the brand construction seem to be necessary to be clearly defined and match the figure 1.

4.3. variables in study 1 and study are not clear and should add more explanation.

Also, I am not sure why the authors attempted to conduct two studies and need to account for the study design.

Conclusions

Resource- technology-, culture-and industry driven modes seem to be analyzed based on external forces. Why are the dimensions ‘brand driving modes’?

Round 2

Reviewer 1 Report

Dear Authors,

I am pleased with the diligent and conscientious work you have done.

It is evident that you have put a great deal of work and heart into your article. I declare no remarks on the submitted version of the article. 

I wish you good health and good luck!

Best 

your reviewer 

Reviewer 3 Report

Thank you for revising the manuscript based on the comments. This paper seems to be improved. However, A minor thing is that this paper should follow the guidelines of the MDPI citation.